# Defining Equinus Foot in Cerebral Palsy

**DOI:** 10.3390/children9070956

**Published:** 2022-06-25

**Authors:** Axel Horsch, Lara Petzinger, Maher Ghandour, Cornelia Putz, Tobias Renkawitz, Marco Götze

**Affiliations:** Department of Orthopedics, Heidelberg University Hospital, 69118 Heidelberg, Germany; axel.horsch@med.uni-heidelberg.de (A.H.); lara.petzinger@med.uni-heidelberg.de (L.P.); mghandourmd@gmail.com (M.G.); cornelia.putz@med.uni-heidelberg.de (C.P.); tobias.renkawitz@med.uni-heidelberg.de (T.R.)

**Keywords:** cerebral palsy, equinus, dorsiflexion, plantarflexion, definition

## Abstract

Background: Equinus foot is the deformity most frequently observed in patients with cerebral palsy (CP). While there is widespread agreement on the treatment of equinus foot, a clear clinical definition has been lacking. Therefore, we conducted this study to evaluate functional changes in gait analysis in relation to maximum possible dorsiflexion (0°, 5°, 10° and 15°) and in two subgroups of CP patients (unilateral and bilateral). Methods: In this retrospective study, CP patients with different degrees of clinically measured maximum dorsiflexion were included. We further subdivided patients into unilaterally and bilaterally affected individuals and also included a healthy control group. All participants underwent a 3D gait analysis. Our goal was to determine the degree of maximum clinical dorsiflexion where the functional changes in range of motion (ROM) and ankle moment and power during gait were most evident. Then, a subgroup analysis was performed according to the affected side. Results: In all, 71 and 84 limbs were analyzed in unilaterally and bilaterally affected subgroups. The clinically 0° dorsiflexion group barely reached a plantigrade position in the 3D gait analysis. Differences in ROM were observed between subgroups. Ankle moment was quite similar between different subgroups but to a lower extent in the unilateral group. All CP patients had reduced ankle power compared to controls. Conclusions: A cutoff value of clinical ≤ 5° dorsiflexion is the recommended value for defining a functionally relevant equinus foot in CP patients.

## 1. Introduction

Equinus foot is the deformity most frequently reported in patients with cerebral palsy (CP) [1]. There are two main types of equinus deformities, and the management approach differs for each type. The first type is known as dynamic equinus, which occurs when the calf muscles become spastic and, in most cases, there is no actual shortening of the gastrosoleus muscle structure yet. Therefore, it is not managed surgically but rather by physical therapy, foot orthosis, casting or botulinum toxin injection i.a. [2]. The second type is referred to as fixed or static equinus with evident contract shortening of the gastrosoleus muscle [3]. In this case, the condition is usually treated surgically by lengthening the affected muscle or its tendon. A wide variety of surgical interventions have been reported in the literature for this type of equinus, and international consensus for treatment has been reached [4,5].

Importantly, proper identification and diagnosis of equinus foot are considered the first step in implementing an appropriate intervention and in reaching a successful outcome [3]. However, there is a clear lack of consensus in the literature on how to properly identify and diagnose equinus foot in CP. A recent survey among 223 orthopedic surgeons highlighted that most of them regularly perform a clinical gait assessment in CP patients they encounter, not specifying if visual or instrumented [6]. However, when it comes to diagnosing equinus foot, their practices differed: 14% rely solely on foot dorsiflexion above plantigrade with an extended knee and neutral hindfoot, and 86% use the “Silfverskjöld test” in addition to the previous approach [6].

In the same context, orthopedic surgeons depend mainly on ankle dorsiflexion to assess equinus; however, there is no clear cutoff value of ankle dorsiflexion to define equinus foot. It is of course difficult to determine such a value, as in CP, equinus must never be considered in isolation; this applies especially for bilateral but also for unilateral CP. The sagittal gait pattern should be identified and described, as the ankle and knee levels are linked by coupling (e.g., the plantar flexion knee extension couple), and treatment should therefore be indicated with caution. Rang’s aphorism that ”a little equinus is better than calcaneus” can usually be agreed on, and individual decisions have to made with each patient [7].

Surprisingly, only a minority of studies report a cutoff value used to define equinus. For example, Horsch et al. [8] defined equinus foot as ≤5° of clinical ankle dorsiflexion in the extended knee. However, most reports assessing the efficacy of different surgical interventions for equinus foot did not report a clear diagnostic criterion [9,10,11,12,13]. This lack of a standard diagnostic criterion further demonstrates why the actual prevalence rate of equinus foot in CP patients has been called into question [14]. In addition, most surgeons rely mainly on the change in the degree of ankle dorsiflexion (pre- vs. postintervention) in defining successful surgery or improved clinical outcomes in CP patients with equinus foot [15,16,17,18,19].

Based on the aforementioned observations, a standard clinical criterion needs to be established to define the basis for managing equinus foot in CP patients surgically or conservatively. Such clinical criteria would be of great importance in determining how patients are selected for treatment based on their functional impairment, and this might change the way we look at this condition entirely - from the real prevalence of equinus, to the choice of the proper intervention to the reliable definition of a successful outcome. Therefore, we conducted this current research, using a gait analysis, in an attempt to reach a clinically relevant criterion to define CP patients with equinus foot who are eligible for surgery from those who can be managed otherwise. We also aimed to investigate whether this criterion would be applicable to different subtypes of patients (those unilaterally and bilaterally affected).

## 2. Materials and Methods

### 2.1. Study Design and Eligibility Criteria

For this study, data from 173 patients diagnosed with CP were retrospectively reviewed. The data in this work were collected in our motion lab in the time period from 2002 to 2021. Only patients with gross motor function system (GMFCS) I-II were included [20]. Other inclusion criteria were fully available clinical examinations and an instrumented 3D gait analysis, including measurements of the kinetic and kinematic parameters. We searched for patients with a limited range of motion (ROM) in the left leg measured in the clinical examination. More specifically, we included patients with a 0°, 5°, 10° or 15° maximum clinical dorsiflexion in the affected ankle joint. It is common to use 5° steps in stating a joint position, as smaller steps are usually not feasible to determine. The ROM was measured in the knee extension. Therefore, we grouped the patients according to their structural shortening of the calf muscles and did not take the additionally underlying amount of spasticity into account. We excluded patients with a GMFCS level III-IV and ankle dorsiflexion >15°; patients with prior soft tissue surgeries (calf, ischios, psoas, etc.) or botox within the past 6 months before gait analysis; patients that had botulinum toxin injections within 6 months prior to gait analysis; and patients that underwent soft tissue lengthening procedures of the lower limbs in general. We also excluded patients with a maximum clinical dorsiflexion of less than 0°, as this is unanimously seen as a functionally impairing equinus foot, and we tried to establish a value of dorsiflexion that marks the functional shift from good to impaired gait.

### 2.2. Study Participants and Measurements

We divided the subjects in two groups depending on the characteristics of CP. One group contained patients diagnosed with unilateral, and another group with bilateral, CP. The instrumented 3D gait analysis (IGA) and the clinical examination were performed by a specialized study nurse and a physiotherapist with years of neuro-orthopedic experience, and angles were assessed by a single examiner using a standard goniometer and the neutral zero method. Markers were applied on the skin according to a standard protocol (Plug in Gait Model) [21]. Three markers were used on each foot to measure the ankle movements. One of them was placed on the lateral malleolus, another one on the calcaneus and a third one being the toe marker, which is attached to the second metatarsal bone. Gait patterns were captured by 16 cameras from a Vicon System (Oxford Metrics, Oxford, UK). The data were captured with a 120 Hz frequency. A Woltring filter was used. Additionally, three force plates (Kistler Instruments, Winterthur, Switzerland) measured the kinetic parameters of the patients. All participants were asked to walk a distance of 7 m several times at their own walking speed. This examination was performed barefoot and without any walking aids. All data captured in the gait analysis were visualized as different diagrams for every joint in the sagittal, frontal and transverse plane. 

### 2.3. Statistical Analysis

Study subjects were recruited from a patient cohort of the neuro-orthopedic department of our University Hospital and our established CP patient register. Data were documented in Microsoft Excel. Additionally, we used MatlabR for a graphic representation of the gait analysis data. Normal values were obtained from a group of 50 age-matched healthy patients without any form of gait pathology. The baseline characteristics of participants were presented as numbers and percentages for categorical/dichotomous variables, and as means and standard deviations (SDs) for continuous variables. 

## 3. Results

### 3.1. Baseline Characteristics of Included Participants

Overall, a total of 155 limbs were analyzed: 71 in the unilateral and 84 in the bilateral group. Meanwhile, 5 patients were excluded from the unilateral groups [GMFC score > 2 (*n* = 2), data gaps in clinical examinations (*n* = 2) and ankle dorsiflexion > 15° (*n* = 1)] and 13 patients were excluded from the bilateral group [GMFCS > 2 (*n* = 6), data gaps in clinical examinations (*n* = 4) and dorsiflexion > 15° (*n* = 3)]. In the unilateral group, 54.9% of the patients were male, 77.5% were GMFCS level I, mean age at the time of IGA was 16.97 (12.13) years, mean weight was 47.16 (21.6) kg and mean height was 151.03 (22.5) cm. Most patients in the bilateral group were also males (56.0%), with GMFCS level II of 72.6% in this group, a mean age of 16 (10.11) years, mean weight of 43.98 (17.12) kg and a mean height of 147.73 (20.64) cm.

### 3.2. Measured Outcomes in the Bilateral CP Group

The results are demonstrated in three different graphs for every group, divided into ROM, ankle moment and ankle power. Each graph shows a different gait pattern according to the clinically measured dorsiflexion. The norm data are visualized with a gray band in all diagrams for better comparison.

In the bilateral group, 27 patients had a limited maximum dorsiflexion with 0° in the clinical exam and 27 patients could reach 5° dorsiflexion. In 22 patients, 10° dorsiflexion was measured and in 8 patients 15° (Figure 1). The graph shows different curves describing the gait pattern during the gait cycle. The curve representing patients with 0° dorsiflexion in the clinical examination did not reach a plantigrade position of the ankle joint in the gait analysis either. These patients walked in plantarflexion without any heel contact during the whole gait cycle. The gaits corresponding to the patients with 5° and 10° dorsiflexion were similar. In comparison to the 0° patients, they could reach a dorsiflexion in the first 15–50% of the gait cycle. Contemplating the plantarflexion in patients with 5°, 10° and 15° after toe-off (60%), the ROM was approximately comparable to a normal gait. Additionally, the curve for the 15° patients followed a nearly normal course with a slight vaulting pattern in the early stance phase. 

The moments of dorsiflexion and plantarflexion show a typical M-Shape with two maxima in every curve (Figure 2). All the curves, regardless of the dorsiflexion measured, were comparable. The first maxima were between 0 and 20% of the gait cycle, which represented the loading response and early midstance [22]. After that, the plantarflexion moment was pathologically reduced, followed by the physiological maxima in 40–60% of the gait cycle. In comparison with the norm data, the second peaks were lower than the peak seen in the gait of the norm group. The plantarflexion moment of the patients with 0° dorsiflexion in the clinical exam was the lowest. 

The generation of power was reduced in all patients in comparison to the norm data (Figure 3). The power generated in the terminal stance to toe-off was lower than usual in every curve, especially in the group consisting of patients with 0° dorsiflexion. In the beginning of the gait cycle, all graphs show an increased absorption of power, followed by increased power during midstance (10–30% of gait cycle) and a reduced power generation before toe-off (40–60% of gait cycle). 

### 3.3. Measured Outcomes in the Unilateral Group

In the unilateral group, the subjects were separated into 35 patients with a 0° dorsiflexion at the clinical examination and 15 patients with 5°. Ten subjects could reach a 10° dorsiflexion and 11 patients showed a 15° dorsiflexion. In patients with unilateral CP, the ROM was quite similar to that in the bilaterally affected group (Figure 4). The limbs from the 0° group barely attained dorsiflexion in the gait analysis; instead, the ankle remained in plantarflexion during most of the gait cycle. The other curves (orange, blue and black) stay in the area of the norm data until the swing phase, where all the legs showed an increased plantarflexion. 

Referring to the ankle moment, the plantarflexion moment in the loading response was also higher in every curve, but in comparison to the bilaterally affected patients to a lower extent (Figure 5). Additionally, the maximum of the plantarflexion moment during 40–60% of the gait cycle was decreased in every group. The highest moment of plantarflexion appeared in patients with 5° dorsiflexion at the clinical examination. 

In patients with unilateral CP, the ankle power was also decreased in all groups during the terminal stance phase to toe-off (Figure 6). Other than in the bilateral group, the absorption-generation pattern of power in the early and midstance was less prominent. Only the 0° patients showed slightly increased power generation in midstance.

## 4. Discussion

Normally, during the stance phase (0–60% of the gait cycle), the greatest degree of dorsiflexion is needed just before lifting the heel in a fully extended knee, in which case the ankle has to be in a dorsiflexed, but perpendicular position for smooth ambulation [23,24]. Yet, there is still controversy related to the proper degree of dorsiflexion that is truly required for this to happen. Consequently, it would seem reasonable to use a normal range of values for defining the physiological gait, rather than to apply a definitive value. Based on the literature, this accepted range of normal ankle dorsiflexion lies between 3° and 15° beyond the perpendicular plane when the knee is fully extended [23,24,25]. 

Equinus foot is described as decreased ankle joint dorsiflexion; although, there is a lack of consensus on the exact definition and diagnostic criteria. While studies widely agree that static ankle joint equinus represents a reduced range of dorsiflexion at the ankle joint, there is no agreement concerning the degree of dorsiflexion reduction needed for this condition to manifest. Of course, different amounts of spasticity can lead to functionally different patterns. In the present work, we examined the differences in ROM, ankle moment and ankle power in different groups of patients based on the degree of maximum clinical ankle dorsiflexion. Here, among patients in the 0° dorsiflexion group, the subgroup of patients with bilateral CP could not reach plantigrade, and the unilateral subgroup barely reached plantigrade in midstance (40%). Thus, we hypothesize that this value would not be the best cutoff value to define a functionally relevant equinus foot in children with CP.

Owing to this lack of consensus, physicians have used a wide variety of restrictions on dorsiflexion for diagnosis [26]. Sobel et al. [27] proposed that patients have less than 0° of dorsiflexion in order to be diagnosed with equinus (i.e., no step beyond plantigrade), while Orendurff et al. [28] recommended a cutoff value of 5°. On the other hand, DiGiovanni et al. [29] proposed a minimum value of less than 10° of dorsiflexion, which is consistent with the need for at least 10° of dorsiflexion to maintain a normal gait and avoid possibly increased forefoot loading throughout locomotion [28,30]. The aforementioned recommendations are consistent with Meyer’s more recent proposal that, instead of basing an equinus diagnosis on a specific range of dorsiflexion motion, a diagnosis should be verified when the reduction in dorsiflexion reaches a magnitude that increases stress on the Achilles tendon and loading on the forefoot [31]. While it would be expected that forefoot pressure rises during locomotion in order to base an equinus diagnosis on a limit of 10° of dorsiflexion, there is no evidence that this would contribute to the development of defects in the foot or lower leg. 

Our results indicate that ankle ROM in children with CP in the unilateral subgroup with 5°, 10° and 15° dorsiflexion was functionally quite similar to that in healthy controls from the beginning of the gait cycle and up to the mid-swing phase (0–75%), typically with more plantarflexion during the end of the cycle as compared to controls. On the other hand, ROM in the bilateral subgroup with 15° dorsiflexion was also similar to that in controls; however, both the 5° and 10° dorsiflexion groups showed reduced dorsiflexion compared to controls during the period from midstance to toe-off (40–60%). 

Notably, the ankle joint moment was quite similar in both the uni- and bilateral CP subgroups during the gait cycle; however, the extent of plantarflexion in the bilateral group was higher. In all dorsiflexion groups (0°, 5°, 10° and 15°), the ankle moment throughout the gait cycle was quite comparable. When compared to healthy controls, all of the dorsiflexion groups revealed a higher plantarflexion moment during the loading response that lowered during midstance and never reached a physiological peak in terminal stance. This was consistent in both uni- and bilateral CP groups. This clearly shows that during gait, function as seen in kinetics is often worse than clinical examination and kinematics would suggest. This might be due to differences in the amount of spasticity of the calf muscles and weakness of the foot levers.

So far, equinus deformity itself has not been considered separately for uni- or bilateral involvement in most of the studies published; although, studies often present one group or the other. In our experience, the two groups present differently, with the amount of foot lever weakness being greater in unilaterally affected patients. Recent studies suggest that a functionally separate approach is beneficial, which is also in line with our experience and this assessment [32]. In terms of ankle power, both groups (uni- and bilateral) in our study had reduced power when compared to healthy controls. Similarly, the gait analysis of both groups revealed that reduced ankle power was evident in the period from terminal stance to toe-off (40–60%) as compared to controls. However, both groups had increased power during the loading phase in early stance as compared to healthy controls. The only difference between the two groups was found in the midstance (10–30%): the bilateral group revealed higher power than controls; while in the unilateral group, power was similar to that of the healthy or typically developing (TD) controls.

In addition, although a cutoff point of 10° may increase forefoot loading during locomotion, Orendurff et al. [28] proposed that a dorsiflexion ≤ 5° should be used for the diagnosis of equinus foot as they found that forefoot pressure was higher in patients with ≤ 5° dorsiflexion than in patients with more than 5° dorsiflexion (*p* < 0.05). This is consistent with our findings, as we noted that the lowest extent of dorsiflexion was observed in the ≤ 5° dorsiflexion group as compared to all other groups (0°, 10°, 15° and healthy controls), especially at the loading and midstance phases of the gait cycle where the ankle is at a perpendicular plane and the knee is normally fully extended. In the same context, ankle moment and power in the ≤ 5° dorsiflexion group were similar to that in other dorsiflexion groups (0°, 10° and 15°) during the whole gait cycle.

Therefore, we recommend taking ≤ 5° dorsiflexion value as a cutoff point for defining equinus foot deformity at the early phases of gait (loading and midstance) when the knee is normally fully extended and the ankle is in a perpendicular position, especially in bilaterally affected CP patients.

Our study provides a clinically relevant criterion for distinguishing CP patients with impaired function due to equinus from those in whom function is not impaired. This criterion is applicable and easy to use in a clinical setting to identify patients who are in need of treatment and those who possibly are not. That being said, this is a simplistic approach to define a functional relevant equinus, and sagittal gait patterns should always be considered before treatment suggestions are made. Surgeons have to be careful not to contribute to, e.g., crouch gait and impairment of global gait function by trying to improve ankle kinematics solely based on the presence of functionally relevant equinus foot.

The main aim of this study was to estimate a clinically relevant criterion to define CP patients with equinus foot, by reaching a cutoff point of dorsiflexion that can help discriminate CP patients with functional impairment who might be in need of surgical management. In the case of instability in the talonavicular joint, the joint was held during the clinical examination, and then the extent of movement was examined. Since it is not possible to avoid the occurrence of a midfoot break during a gait analysis, this should be considered a clear limitation. However, the purpose of this study was to find a clinical definition for equinus foot with the help of a gait analysis. Our patients were first clinically examined, which ensured that no midfoot break was present. It would also be worth discussing the integration of a foot model in future studies in order to investigate this question more closely. 

Our study has several limitations. The most important limitation is the small number of participants within each subgroup (0°, 5°, 10° and 15°) included in our analysis, which further limits the implications drawn from our study and renders it difficult to apply our findings more generally. Additionally, the retrospective nature of our study limits the interpretation of our results. Moreover, we describe the differences observed between the groups studied clinically; thus, more work is required to determine whether these minimal clinical differences also reach statistical significance. Therefore, and based on the aforementioned limitations, future research in this area is still warranted by larger studies with more diverse populations and consideration of sagittal gait patterns to be able to determine whether such definition criteria are applicable for all CP patients, regardless of their clinical and demographic characteristics.

## 5. Conclusions

In conclusion, a cutoff value ≤ 5° maximum clinical dorsiflexion should be used to distinguish between patients who might need treatment from those who do not among CP patients with equinus foot. However, the applicability of this criterion among patients who are unilaterally and bilaterally affected is slightly different; therefore, we recommend considering the laterality of CP when diagnosing patients with equinus foot and when choosing the appropriate management approach for these individuals. This simplistic approach should not leave the sagittal gait pattern unconsidered, as this might lead to impairment of the global gait function.

## Figures and Tables

**Figure 1 children-09-00956-f001:**
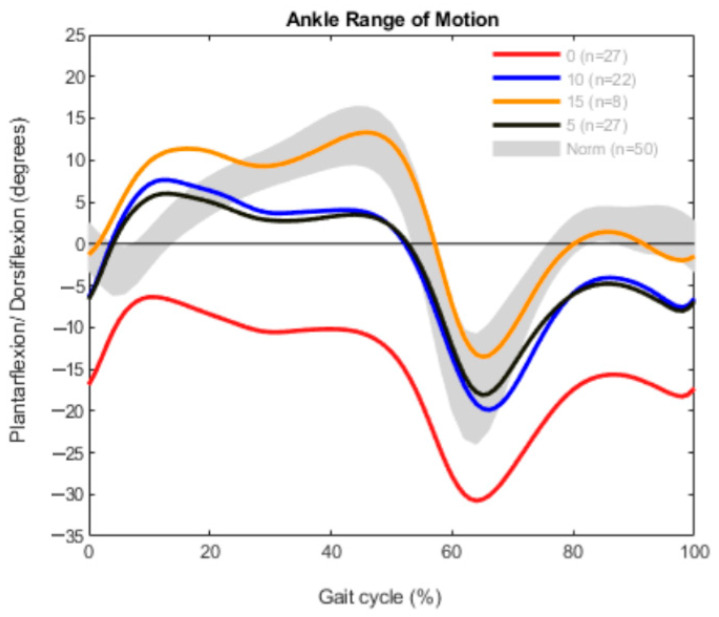
Range of motion measurement in the bilateral group (*n* = 84).

**Figure 2 children-09-00956-f002:**
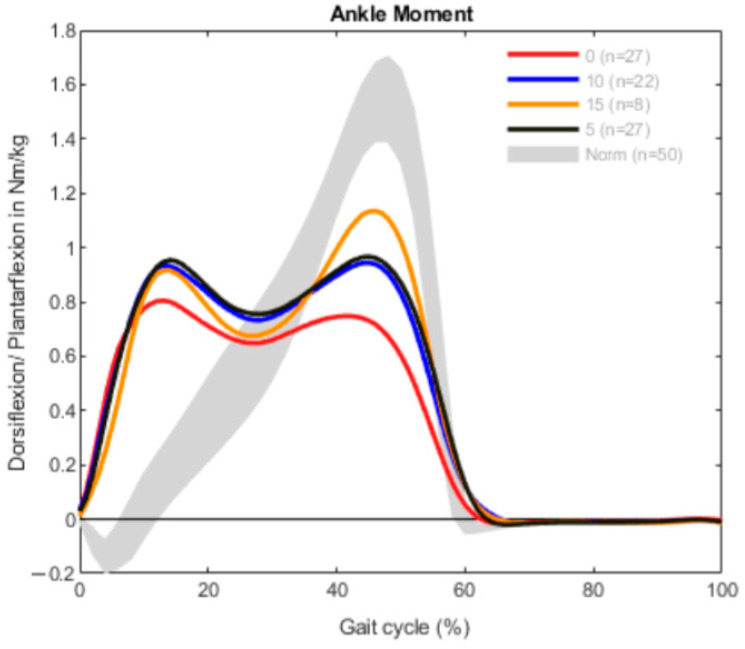
Ankle moment measurement in the bilateral group (*n* = 84).

**Figure 3 children-09-00956-f003:**
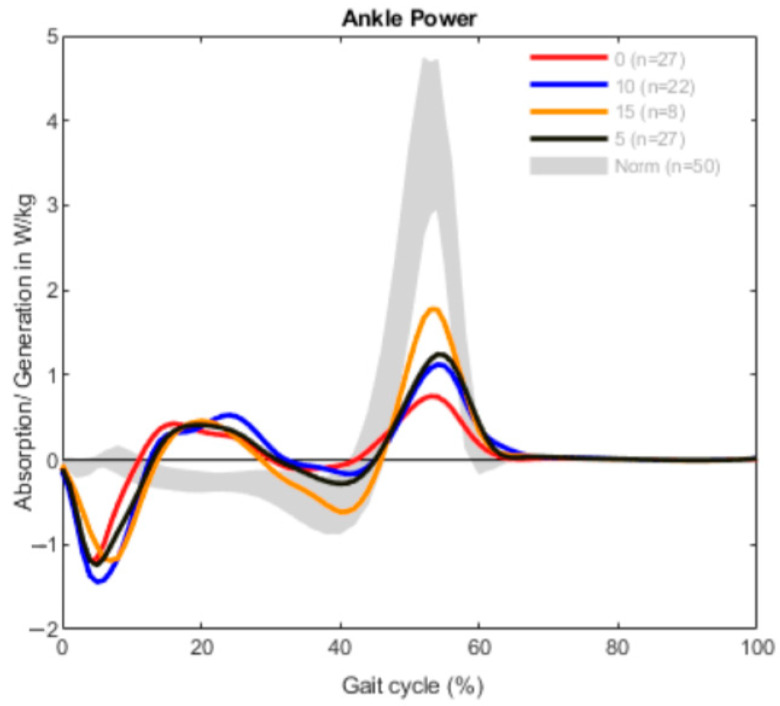
Ankle power measurement in the bilateral group (*n* = 84).

**Figure 4 children-09-00956-f004:**
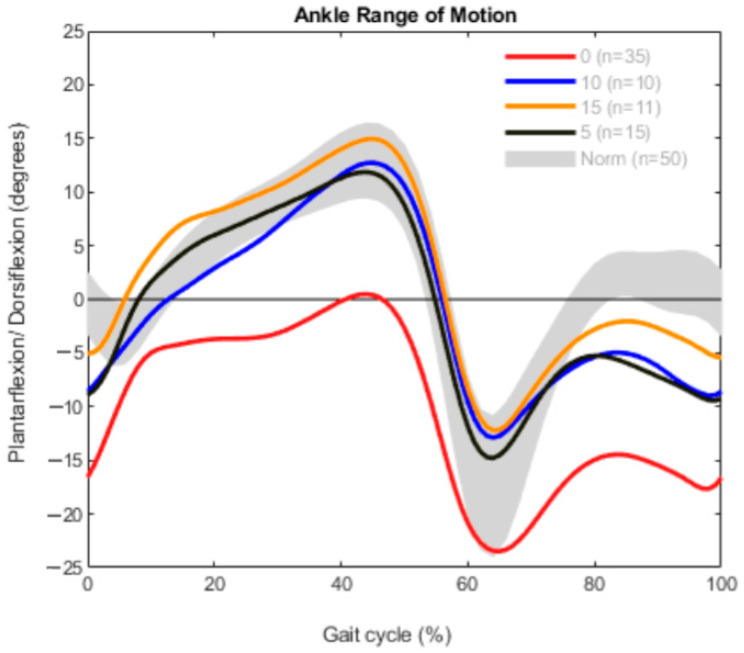
Range of motion measurement in the unilateral group (*n* = 71).

**Figure 5 children-09-00956-f005:**
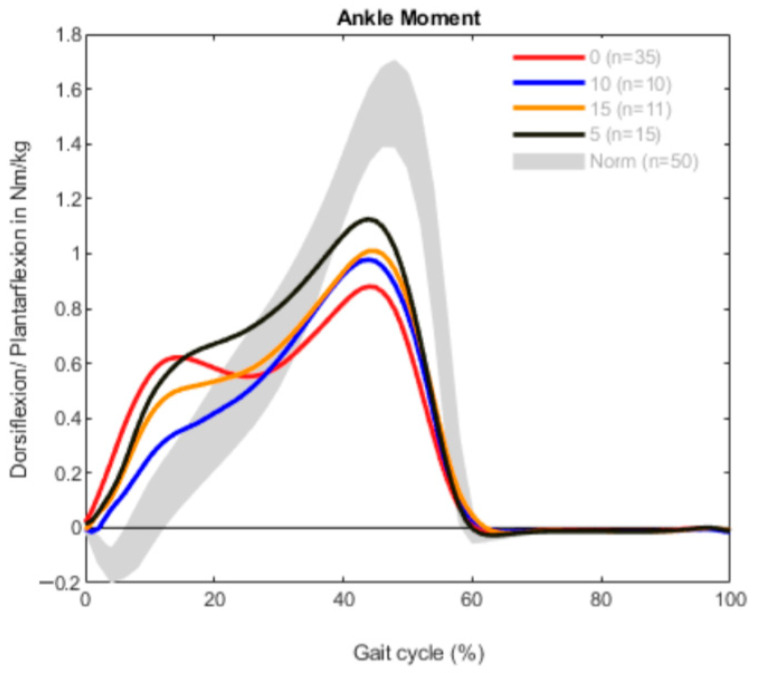
Ankle moment measurement in the unilateral group (*n* = 71).

**Figure 6 children-09-00956-f006:**
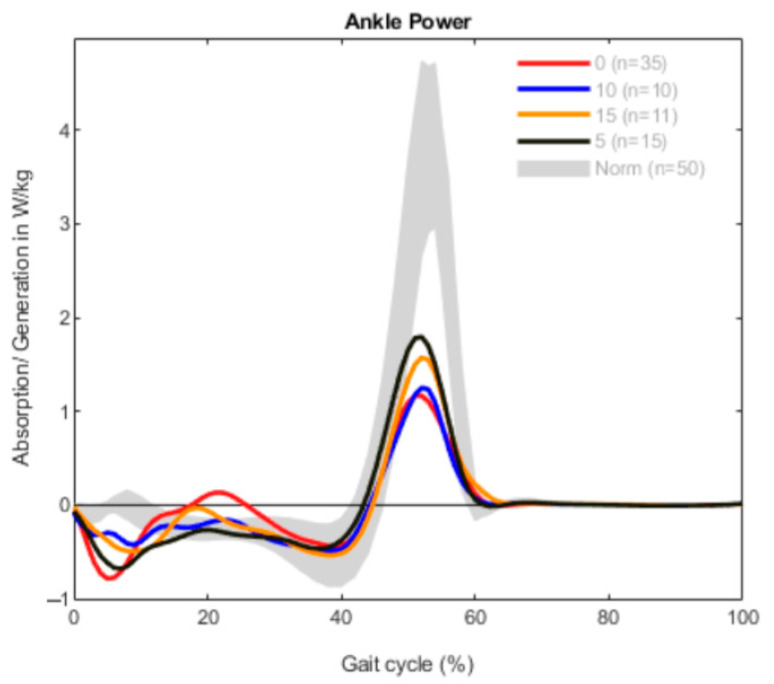
Ankle power measurement in the unilateral group (*n* = 71).

## Data Availability

All the Data analyzed in this manuscript can be provided upon request by contacting the corresponding author.

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
