# Peer review of "Defining Equinus Foot in Cerebral Palsy"

_children, 2022, doi:10.3390/children9070956_

Round 1
Reviewer 1 Report
It is not clear whether all the children in this study were scheduled to undergo surgery. It is questionable whether it would be meaningful to divide groups and conduct gait analysis for children who do not need surgery. Further I don't think it's appropriate in scientific papers to use angles in increments of 5 degrees. Measurement error may be large. Criteria of data gap between the two examiners are ambiguous..
I don't understand why authors came to the conclusion that a cutoff value of clinical <5 degree dorsiflexion is the recommended value for defining a functionally relevant equinus foot in CP patients.
Author Response
Thank you for your comment.
Among other things, the need for surgery was decided on the basis of the gait analysis. in the clinical examination, it is difficult to implement a degree specification that is exact to one degree, so it is not unusual to measure in 5° steps. Since our proposed definition is a clinical definition, we consider this to be a feasible approach. If the gait curves of the different degrees of movement are examined more closely, it is noticeable that patients who clinically measured have a dorsal extension of 0° and do not have any heel contact in the gait analepsis. Patients with 5° dorsal extension have heel contact and "still tolerable" gait parameters. this fact leads us to our conclusion.
Reviewer 2 Report
The paper presented for the review gives the interesting information regarding biomechanical nuances of equinus gait in children with cerebral palsy. The authors analysed 2 groups of children - with unilateral and bilateral cerebral palsy. Based on the data obtained thy proposed categorical definition of equinus which can be assessed clinically and confirmed by the goniometric assessment. The paper provides practically significant information, clearly designed and written. It is recommended for publication in the authors’ version.Author Response
No comments were raised. Thank you for your comment and for your valuable feedback.
Reviewer 3 Report
a well structured and executed research with the conclusion well supported by data gathered.
A minor writing error: line 249 "the two groups present different" better to say "present differently"
Author Response
Thank you for your valuable comments and contribution. As for the small edit in line 249, it was corrected as suggested. Many thanks.
Round 2
Reviewer 1 Report
If all the other reviewer's agrees to this paper's significance.
I have no objection
I still have a question how you decided angles as you mentioned two evaluator had participated in measuring.
Author Response
Thank you for bringing this mistake to our attention. We declare that this was a mistake on our part; measurements done in this project were carried out by a single examiner, not two.
The main writer of this manuscript unintentionally got this point mixed up with another project where we had developed an equinometer and compared it to the standard goniometer in measuring the degree of equinus deformity in CP children, where we had 2 examiners conducting measurements.
We greatly appreciate your help in noticing this hardly noticeable mistake and in significantly improving the quality of our manuscript. Thank y ou!
This manuscript is a resubmission of an earlier submission. The following is a list of the peer review reports and author responses from that submission.
Round 1
Reviewer 1 Report
The authors present a paper looking at equinus deformity in cerebral palsy. In its current format I have a number of concerns regarding the article and will not be recommending it for publication.
The title “ Which patients with cerebral palsy and equinus foot benefit from surgery?”. This paper does not in anyway evaluate surgical outcomes, or mention surgical status of the included participants. This title is misleading.
The authors are looking at equinus foot deformity, they only selected participants that had 15, 10, 5, 0 degrees of dorsiflexion measured in knee extension. Also why are measures only in 5 degree increments are all measures rounded? Does this mean they excluded participants with considerable equinus, ie those that actually had plantarflexion. Nothing in the methods is written about differentiating dynamic equinus from fixed equinus.
The statistical analyse includes information that is not analyses and there is no actual analysis of the data in this article. Gait traces should have sd for groups not just norms.
The methods lack detail in measurements there is no mention of physical examination measures. Data select is unclear and if those with previous surgery are included in this study. Information about processing of 3DGA data needs to be included, ie filtering capture frequency ect.
Author Response
We would like to thank the reviewers for their feedback and valuable input. Please find our responses below:
The authors present a paper looking at equinus deformity in cerebral palsy. In its current format I have a number of concerns regarding the article and will not be recommending it for publication.
Authors:
We would like to thank you for this valuable input, that made us overthink a few points.
The title “Which patients with cerebral palsy and equinus foot benefit from surgery?”. This paper does not in anyway evaluate surgical outcomes, or mention surgical status of the included participants. This title is misleading.
Authors:
We absolutely agree with changing the title of the work back to a title we initially had in mind and would like to propose the following:
“Defining Equinus Foot in Cerebral Palsy”.
The authors are looking at equinus foot deformity, they only selected participants that had 15, 10, 5, 0 degrees of dorsiflexion measured in knee extension. Also why are measures only in 5 degree increments are all measures rounded? Does this mean they excluded participants with considerable equinus, ie those that actually had plantarflexion. Nothing in the methods is written about differentiating dynamic equinus from fixed equinus.
Authors:
Measures are in 5 degree increments because in our gait laboratory it is common to document the clinical examination using 5 degree steps. Recording the dorsiflexion in smaller steps in between is in our opinion neither reasonable nor useful. If the patient has 0 degree dorsiflexion in the clinical examination it means the patient is not able to perform dorsiflexion with the knee extended at all. We didn´t exclude patients with less than 0 degrees dorsiflexion rather than choosing patients based on their dorsiflexion to find a clinical cutoff value defining a pes equinus as soon as it is functionally relevant and that is easy to access in clinical examination. Based on the clinical (and not functional) data we used in this study a differentiation in dynamic or fixed equinus is not possible because the range of motion is evaluated by passive examination, hence all values show the actual structural shortening of the tendon.
The statistical analyse includes information that is not analyses and there is no actual analysis of the data in this article. Gait traces should have sd for groups not just norms.
Authors:
We did only graphically illustrate the gait data with a specific software to visually show the traces without having calculated standard deviations.
The methods lack detail in measurements there is no mention of physical examination measures. Data select is unclear and if those with previous surgery are included in this study. Information about processing of 3DGA data needs to be included, ie filtering capture frequency ect.
Authors:
We spared this very specific information on purpose, as it is of no value to the readers. We nonetheless added a sentence to the manuscript in paragraph 2.2:
Reviewer 2 Report
The paper presented for review is aimed to formulate clear clinical definition of equinus foot.
The authors conducted meticulous study of different modalities of gait biomechanics in children with unilateral and bilateral cerebral palsy which have limited dorsiflexion of the foot. The term “pes equinus” came from the historical descriptive terminology but stilll has conceptual meaning for the complex understanding of normal walking and gait abnormalities. For the practical reason the consensus definition of equinus is important for diagnostic and treatment purposes. The authors propose the cutoff amount of passive foot dorsiflexion for the determining of equinus based on kinematics and kinetics of the lower limbs during gait. The fact that the study was conducted independently in children with hemiplegic and diplegic cerebral palsy is important for the clinical reasons and makes the results even more valuable.
To the reviewer’s knowledge this is the most representative study regarding this theoretically and practically meaningful topic.
Author Response
We would like to thank the reviewers for their feedback and valuable input. Please find our responses below.
The paper presented for review is aimed to formulate clear clinical definition of equinus foot.
The authors conducted meticulous study of different modalities of gait biomechanics in children with unilateral and bilateral cerebral palsy which have limited dorsiflexion of the foot. The term “pes equinus” came from the historical descriptive terminology but stilll has conceptual meaning for the complex understanding of normal walking and gait abnormalities. For the practical reason the consensus definition of equinus is important for diagnostic and treatment purposes. The authors propose the cutoff amount of passive foot dorsiflexion for the determining of equinus based on kinematics and kinetics of the lower limbs during gait. The fact that the study was conducted independently in children with hemiplegic and diplegic cerebral palsy is important for the clinical reasons and makes the results even more valuable.
To the reviewer’s knowledge this is the most representative study regarding this theoretically and practically meaningful topic.
Authors:
Thank you very much for agreeing with us on the importance of this topic and the value of this work.
Reviewer 3 Report
The authors carried out a detailed gait analysis to define a clinically important entity of equinus. The methodology is sound, the manuscript is well written and the conclusion well supported by the data garnered.
Author Response
We would like to thank the reviewers for their feedback and valuable input. Please find our responses below.
The authors carried out a detailed gait analysis to define a clinically important entity of equinus. The methodology is sound, the manuscript is well written and the conclusion well supported by the data garnered.
Authors:
Thank you for your feedback and support.
Round 2
Reviewer 1 Report
The authors falsely assume that changes to the ankle dorsiflexion kinematics from normal are solely the result of a static equinus deformity and completely ignore that this is likely the result of both calf contracture and spasticity. If these children underwent a physical examination of would assume measures of calf muscle spasticity were assessed also. It is unclear why this information is not included, as it could substantially alter the conclusions of the paper.
Also this paper is descriptive of the gait changes, no statistical analysis is performed but from this the authors are conclude a hard cut-off for determining suitability for surgery based on a single clinical examination measure of ankle range of motion. Not only is this conclusion lacking the data to support it, it is dangerous recommendation.
I have a number of other concerns regarding this manuscript outline below in detail.
In response to the authors comments
The authors have clarified that clinical examination measures are rounded to 5 degree increments at the time of recording. However, no modification to the text was made to make this clear to the reader.
The authors state “We didn´t exclude patients with less than 0 degrees dorsiflexion rather than choosing patients based on their dorsiflexion to find a clinical cutoff value defining a pes equinus as soon as it is functionally relevant and that is easy to access in clinical examination.”
This statement is unclear to me, were children with less than 0 degrees dorsiflexion (ie in plantarflexion) included in your study? Secondly if you are looking to find a cut-off those below 0 need to also be included, otherwise you are already assuming a cut-off of 0 degrees prior to conducting the study.
“(since we have taken data from the clinical examination, it is not 79 possible to distinguish between dynamic and fixed equinus)..”
The methods lack detail in measurements there is no mention of physical examination measures. Data select is unclear and if those with previous surgery are included in this study. Information about processing of 3DGA data needs to be included, ie filtering capture frequency ect.
Authors:
We spared this very specific information on purpose, as it is of no value to the readers. We nonetheless added a sentence to the manuscript in paragraph 2.2:
Introduction
Pg 1, line 39
“there is a clear lack of consensus in the literature on how to properly 39 identify and diagnose equinus foot in CP.” I think the authors are confusing the fact that the decision to perform calf lengthening surgery for equinus is multifactorial not based solely on a single measure.
Line 40
“ A recent survey among 223 orthopedic surgeons highlighted that most of them perform clinical gait assessment in all CP patients they encounter [5].”
This is a misinterpretation of the results from this study. Firstly it identified that most paediatric orthopaedic surgeons regularly\frequently perform a clinical gait assessment, when considering calf lengthening surgery, secondly the survey in question does not specify if the gait assessment was visual gait analysis or instrumented.
Methods
“Three markers are used on each foot to measure the ankle movements.” This statement is poorly worded. Markers are required on shank and foot to measure foot movement. Malleolus is not on the foot but on the tibia. I think you need to include in your limitations the toe marker being so far down the foot may lead to inaccurate ankle dorsiflexion measurements in children with mid-foot breach which is common in children with CP.
While the authors may deem these details about data capture, that do not require much text unnecessary I disagree. I tis common practise in gait analysis papers to include this level of detail and allows for transparent and reproducible science.
“The gait pattern of the subjects was captured by 16 cameras from a Vicon System (Oxford Metrics, Oxford, UK). The data was captured with a 120Hz frequency using a Woltring filter. Additionally, three force plates (Kistler Instruments, Winterthur, Switzerland) measured the kinetic parameters of the patients.”
This needs to be reworded, in its current format it does not appear that the authors are familiar with how the motion capture system functions. Data is not capture with a filter?? Additionally while you added marker data capture rate, 120Hz, you choose to omit this for forceplate data which is generally 10 times that of the marker data.
Still no statistical analysis to identify cut-ff, still purely subjective by authors.
Results
Gait traces, I assume mean and standard deviation is included for normal group, although not stated in figure capture what the grey band represents. Why do the different dorsiflexion groups not have sd band as well, is that because there was a lot of variation?? Would be easier for readers if key went 0,5,10,15 ie natural number not computer numbering 0,10,15,5
I think an interesting point that is not mentioned from the gait analysis is that while the 15 degree group (bilateral), 5,10,15 group (unilateral) achieved relatively normal kinematics they were unable to achieve normal kinetics, why is this?
Another point worth discussing is why if they can achieve 0 degree dorsiflexion on clinical examination, why can they not during gait, spasticity? This is why you should include measures of spasticity in your study.
Discussion
“So far equinus deformity has not been considered separately for uni- or bilateral in- volvement in most of the studies published.”
Studies often only present on either bilateral or unilateral CP. It has long be recognised that they present differently with equinus and management is different.